# Thermal Barrier Stability and Wear Behavior of CVD Deposited Aluminide Coatings for MAR 247 Nickel Superalloy

**DOI:** 10.3390/ma13173863

**Published:** 2020-09-01

**Authors:** Dominik Kukla, Mateusz Kopec, Zbigniew L. Kowalewski, Denis J. Politis, Stanisław Jóźwiak, Cezary Senderowski

**Affiliations:** 1Institute of Fundamental Technological Research, Polish Academy of Sciences, Pawińskiego 5B, 02-106 Warsaw, Poland; mkopec@ippt.pan.pl (M.K.); zkowalew@ippt.pan.pl (Z.L.K.); 2Department of Mechanical and Manufacturing Engineering, University of Cyprus, 20537 Nicosia, Cyprus; politis.denis@ucy.ac.cy; 3Faculty of Advanced Technologies and Chemistry, Military University of Technology, 00-908 Warsaw, Poland; stanislaw.jozwiak@wat.edu.pl; 4Department of Materials and Machinery Technology, University of Warmia and Mazury, Oczapowskiego 11 St., 10-719 Olsztyn, Poland; cezary.senderowski@uwm.edu.pl

**Keywords:** chemical vapor deposition, nickel alloys, coatings, X-ray analysis

## Abstract

In this paper, aluminide coatings of various thicknesses and microstructural uniformity obtained using chemical vapor deposition (CVD) were studied in detail. The optimized CVD process parameters of 1040 °C for 12 h in a protective hydrogen atmosphere enabled the production of high density and porosity-free aluminide coatings. These coatings were characterized by beneficial mechanical features including thermal stability, wear resistance and good adhesion strength to MAR 247 nickel superalloy substrate. The microstructure of the coating was characterized through scanning electron microscopy (SEM), X-ray energy dispersive spectroscopy (EDS) and X-ray diffraction (XRD) analysis. Mechanical properties and wear resistance of aluminide coatings were examined using microhardness, scratch test and standardized wear tests, respectively. Intermetallic phases from the Ni-Al system at specific thicknesses (20–30 µm), and the chemical and phase composition were successfully evaluated at optimized CVD process parameters. The optimization of the CVD process was verified to offer high performance coating properties including improved heat, adhesion and abrasion resistance.

## 1. Introduction

Nickel alloys used for the construction of aircraft engines are characterized by high performance properties including corrosion, heat and creep resistance [1,2]. Additional aluminide coatings deposited on these alloys improve their thermal and chemical resistance during operation in high temperature and aggressive environments [3,4]. Specifically, aluminide thermal coatings have been found to effectively prevent oxidation and carbonization in high temperature conditions [5,6].

Increasing demands by the aircraft industry have resulted in improved engine efficiency and high operating temperatures [7,8], which has resulted in increased use of nickel alloys. Turbine components made from nickel alloys are exposed to extremely high temperatures, as the gas temperature upstream of the turbine may exceed 1650 °C [9]. In order to increase the operational life of turbine blades and other engine components, protective layers are commonly used. Thermal barrier coatings/bond coating (TBC/BC), with a complex chemical composition and structure, are currently used for turbine blades made of nickel alloys [10]. The outer ceramic layer is designed to reduce the thermal and erosive effects during take-offs and landings. Moreover, low thermal conductivity enables the elimination of rapid temperature changes and thermal expansion effects at different stages of engine operation. The bonding interlayer (bond coating, BC) is designed to protect engine components from oxidation and subsequently increases the adhesion of the ceramic layer to the substrate material. Such coatings are currently produced by plasma spraying (PS), electron-beam physical vapor deposition (EB PVD) and gas detonation spraying (GDS) techniques [11,12]. In plasma spraying, a powder of a carefully selected chemical composition is used to obtain the protective coating. In gas detonation spraying (GDS), FeAl intermetallic coatings obtained from self-decomposing powders are characterized by improved mechanical properties, such as hardness, thermal resistance and adhesive strength of the coating/substrate bond. The physical-chemical properties of the GDS protective interlayers with compact and lamellar structure could enhance the properties of the coating systems. It is possible to tailor the substrate coating by applying multiple structures and properties at individual areas of a workpiece, and thus control the influence of temperature and stress gradients. The application of an intermediate layer could improve the adhesion of GDS coatings through the enhancement of cohesive strength on grain boundaries. Moreover, the refinement and homogenization of the coating consisting of equiaxed, fine grains (<1 µm) can result in the formation of a multiphase, refractory intermetallic-based thermal barrier reinforced with Al_2_O_3_ oxides. Such intermediate layers could improve the adhesion between the interlayer and substrate by over 50 MPa. The complex microstructures of these layers also exhibit a uniform distribution of compressive stress throughout their volume, which has been confirmed by the Sachs–Davidenkov method [13]. Uniform stress distribution is typical for GDS processes, where there is limited thermal interaction between the detonation products and substrate. It may be concluded that the deposition of intermetallic coatings leads to thermal stress generation (due to differing linear thermal expansion coefficients), which can influence the integrity and durability of the coating/substrate bond system.

Therefore, the deposition of aluminum-based coatings must be performed with optimized process parameters to ensure the quality of the coating. Aluminide thermal coatings promise to reduce the sliding wear of turbine blades, and subsequently, increase the efficiency and lifetime of the engine. Taking into account the potential benefits for intermediate layer applications, the aim of this work was to determine the microstructural and mechanical properties of aluminide coatings deposited on a MAR 247 nickel superalloy using optimized chemical vapor deposition (CVD) process conditions in order to obtain a stable thermal barrier and wear resistant coating with satisfactory bond strength.

## 2. Materials and Methods

MAR 247 nickel superalloy was produced by using the casting process with uniform crystallization performed in ceramic molds. Sample casting was performed in an ALD vacuum furnace. Samples made of MAR 247 alloy with directional grain orientation (Figure 1a) were transferred out of the furnace at the controlled speed of 3 mm/min. Samples with equiaxed microstructure (EQ) were quenched in the furnace to achieve the required microstructure. The chemical composition of MAR 247 nickel superalloy is presented in Table 1.

Aluminide coatings were produced in the process of low activity aluminization by chemical vapor deposition (CVD) using an Ion-Bond setup (Figure 1b) (Ion Bond Bernex BPX Pro 325 S, IHI Ionbond AG, Olten, Switzerland) located in the Materials Testing Laboratory of the Rzeszów University of Technology. The CVD processes were performed on 50 samples of MAR 247 alloy in nitrogen/hydrogen protective atmospheres for 1–12 h at the temperature of 1040 °C and internal pressure of 150 mbar. The microstructural characterization and chemical composition analysis of the coatings were examined using a HITACHI SU70 scanning electron microscope (Hitachi, Tokyo, Japan) with energy dispersive spectroscopy attachment (EDS, by Oxford Instruments, Oxford, UK) and HITACHI 260 also with an EDS detector. Phase composition was evaluated using CuKα radiation with a Bruker D8 X-ray diffractometer (Bruker, Billerica, MA, USA) and Rigaku (Tokyo, Japan) Ultima IV diffractometer with Co-K radiation (λ¼1.78897 Å) and operating parameters of 40 mA and 40 kV with a scanning speed of 1°/min and a scanning step of 0.02° in the range of 20°–150°. The microhardness of coatings was determined on a ZWICK hardness tester (Materialprüfung 3212002, Ulm, Germany) using the Vickers method. The sliding wear test was performed using the three rollers + cone method in accordance with the Polish Standard (PN-83/H04302). The tests were performed with the counter cone-sample made from SW7M high-speed steel under a constant pressure of 200 MPa and controlled time of 100 min. The adhesion of the coatings was assessed by applying a Micro-Combi-Tester (MCT^3^, Anton Paar, Warsaw, Poland), with the force increasing from 0 to 100 N.

## 3. Results and Discussion

### 3.1. Microstructural Characterization of Coatings after CVD Process

The test matrix of the CVD parameters used in the study is presented in Table 2. The parameters include a temperature of 1040 °C with differing deposition times ranging from 1–12 h to assess the maximum effectiveness of the process. These parameters were selected on the basis of preliminary studies performed at different temperatures from the range of 880 °C to 1040 °C. In these tests, it was found that CVD performed at lower temperatures (880 °C, 950 °C) with a relatively long deposition time (up to 12 h) and with different protective atmospheres failed to obtain a defect free and cohesive coating, as presented in the sample cross-sections of Figure 2.

The deposition process performed at the initial temperature of 880 °C for 12 h under nitrogen protective atmosphere resulted in an incoherent coating with visible cracks along the surface (Figure 2a). Subsequent increase in the process temperature to 1040 °C led to excessive deterioration of the coating as shown in Figure 2b. Based on such observations, it was concluded that extension of the deposition time for a nitrogen protective atmosphere would have yielded unsatisfactory results. Therefore, similar process conditions were evaluated for the hydrogen atmosphere. Experimentation showed that at temperatures lower than 950 °C, defects such as cracks occurred within the structure (Figure 2c). Based on the microstructural observations, it was found that the optimal conditions include a temperature of 1040 °C with hydrogen as the protective gas atmosphere to perform the CVD process. These variables demonstrated the formation of a successful aluminide coating (Figure 2d), and thus, the process variable that was further studied to improve coating quality was deposition time.

The coating quality was evaluated by evaluating the homogenous morphology as well as the intermetallic behaviour. Figure 3 shows the surface morphology evolution of the deposited coatings depending on CVD deposition time with temperature and protective atmosphere maintained as constant. Aluminide coatings obtained during 1 h deposition time (Figure 3a,b) were characterized by ~10 µm radius craters and microvoids observed over the entire surface. The occurrence of these voids, as well as sharp-like grains in the highly developed cellular coating structures (Figure 3b), may indicate that the deposition time was too short to attain a protective coating without defects. After the deposition time was increased to 2 h (Figure 3c,d), the size of the craters slightly increased and a smooth structure was achieved. Despite the longer time of deposition, a porous structure was still observed (Figure 3d). A considerable change of the morphology was obtained when the deposition time was extended up to 8 h (Figure 3e–h). The coating deposited on MAR 247 (Figure 4a) after 12 h was characterized by a homogenous, non-defect structure (Figure 3g,h). Large, NiAl intermetallic crystallites were observed on the surface of the layer and their composition was confirmed by X-ray phase analysis (Figure 4b). The first peak observed on X-ray diffraction patterns (NiAl (100)) may indicate that formation of NiAl intermetallic superstructure (secondary solid solution β with B2 ordered structure) occurred. Such structure is stable at temperatures up to approximately 1400 °C with a wide range of aluminum content (31 at.% to 58 at.%) [14]. The chemical composition of this specific coating contained approximately 42.55% aluminum and 51.51% nickel, which may indicate that the NiAl was the dominant phase (Table 3).

Figure 5 presents an example of the intermetallic coating produced during the process of low-active high temperature aluminization (LAHT) at the temperature of 1040 °C for 12 h with additional distribution of main elements. This coating consists of two main sublayers. A chemical composition tested on the cross-section in micro-areas showed that the outer sublayer with a thickness of approximately 35 µm contains mainly aluminum (33.2 at.%) and nickel (55.4 at.%). In the inner sublayer, chromium (18.7 at.%) and molybdenum (9.96 at.%) were additionally observed. It was concluded that the NiAl layer limits chromium inter-diffusion, which may explain its low content in the outer sublayer and its increased concentration in the inner sublayer. The solubility of this element in the NiAl phase is limited to several percent. The EDS map analysis (Figure 5) also confirmed the assumptions of LAHT aluminization (900 °C–1150 °C) reported in literature [15]. In this process, the aluminum is deposited on the surface at a reduced rate. This allows nickel atoms to simultaneously diffuse outward to the surface and form a β-NiAl surface layer. The high content of nickel atoms observed in the coating area (Figure 5) was associated with the low aluminum activity at the surface, which effectively holds the surface aluminum content close to 50 at.%.

Microstructural characterization and X-ray analysis allowed for the optimization of chemical vapor deposition (CVD) parameters of MAR 247 nickel superalloy. The CVD process performed at 1040 °C for 12 h in protective hydrogen atmosphere enabled a non-defect coating with homogenous structure and uniform thickness to be obtained (Figure 6).

### 3.2. Microhardness Profiles

Figure 7 shows the microhardness profile of aluminide coating. An evolution of the microhardness was captured from the edge of the aluminized sample to its core. It was clearly observed that the coating exhibits an improved microhardness in comparison to the substrate. The coating was characterized by hardness of 664 HV0.05, which was associated with the distribution of the aluminum atoms in the lattice of NiAl intermetallic coating. The enhancement in hardness could be also ascribed to the formation of NiAl phases during high temperature deposition. The hardness of the material gradually decreased from the edge to core of the material and maintained at the stable value approximately equal to 460 HV0.05. The maximum hardness, greater than 650 HV0.05, occurred in the coating area where Cr, Mo, Co carbides were dominant. The occurrence of such carbides reinforces the solution strengthening, and thus, improves the hardness. Furthermore, a high temperature promotes diffusion processes and reaction kinetics in the substrate/coating region, leading to the smooth hardness transition between coating areas, as shown in Figure 7. The microhardness of the transition zone was less than that of the aluminized coating surface because elements of the substrate (such as chromium and molybdenum) were excessively diffused during the process of solidification. Hardness in the transition area was approximately equal to 550 HV0.05, while that of the core of MAR 247 slightly increased to approximately 440 HV0.05. This is presumably caused by the relatively high temperature that the CVD process was executed in.

### 3.3. Heat Resistance Properties of MAR 247 Alloy with Aluminide Protective Coating

As discussed in the introduction, modern aircraft engine turbines operate at high temperatures, which limits the materials that can be used. Contemporary nickel superalloys are limited to maximum operating temperatures of 1100 °C, which necessitates the use of protective coatings. Therefore, to replicate these operating conditions, heat resistance tests were performed on MAR 247 alloy with an aluminide protective coating in an air atmosphere at 1100 °C for 24 h. 

The morphology of MAR 247 nickel superalloy in the initial state (without coating) (a), and with the NiAl diffusive intermetallic coating (b), are illustrated in Figure 8. X-ray phase analysis of the oxidized MAR 247 alloy without coating revealed that its phase structure contains mostly nickel oxide (NiO) (Figure 8a and Figure 9a). On the contrary, the NiAl coated MAR 247 nickel superalloy exhibited no changes in the layer morphology. Subsequent SEM analysis revealed the fine-crystalline particles formed on the surface (Figure 8b). The phase structure of scale consisted mainly of NiAl and NiAl_2_O_4_ intermetallic phases as well as stable, alumina oxide (α-Al_2_O_3_) (Figure 9b). It was found that a high concentration of aluminum atoms near the coating surface area (Figure 5) allows for the formation of a thick, protective alumina oxide (α-Al_2_O_3_)-based scale and led to the improvement in hot corrosion resistance of the MAR 247 nickel superalloy. The aluminide coating subjected to oxidation was characterized by the excellent durability and tightness of the protective scale as no scale spallation was observed (Figure 8b).

### 3.4. Characterization of Adhesion and Wear Resistance of MAR 247 Alloy with Aluminide Protective Coating

In order to assess the adhesion of the coatings, scratch tests were performed under a critical force of 100 N and 10 mm displacement. Despite the number of conformal cracks in the perpendicular to scratch direction, neither spallation nor a breakdown of the layer was observed on the coating surface, as shown in Figure 10. It should be noted that the aluminide coatings exhibited a very good adherence.

The results of wear resistance tests on the uncoated and coated MAR 247 alloy were presented in Figure 11. The uncoated MAR 247 alloy was characterized by the linear wear of 22 µm after 100 min of friction, which is twice as large as that for the coated alloy. A significant wear increase in the uncoated MAR 247 alloy can be easily observed after approximately 30 min. It was noted that after a relatively long time of friction (100 min), the NiAl coating remains firmly adhered to the substrate. The microstructural assessment of worn tracks (Figure 12) suggests the formation of third-body particles, associated with the observed scoring and grooving shallow marks. The presence of third-body particles in sliding systems is widely reported in the literature [16,17], and is related to the detachment of material from the surfaces being in the contact during sliding.

## 4. Conclusions

Chemical vapor deposition for MAR 247 nickel superalloy was performed at a temperature of 1040 °C for 12 h in a protective hydrogen atmosphere. This process was demonstrated to produce a non-defect substrate material with thermal barrier and wear resistant uniform thickness NiAl coating from 20 µm to 30 µm. The coating was characterized by the very good adherence, wear and thermal resistance confirmed by experimental studies. Its application improved the mechanical properties, such as hardness and wear, by almost a factor of two compared to the as-received MAR 247 alloy. The coating hardness of 664 HV0.05 was associated with the distribution of aluminum atoms in the lattice of the NiAl intermetallic coating formed during high temperature deposition. The aluminide coatings exhibited a very good adherence during scratch tests as the breakdown of the layer was not observed on the coating surface after testing. The oxidized aluminide coating was characterized by the excellent durability and tightness of the protective scale, as no scale spallation was observed. The phase structure of the scale consisted mainly of NiAl and NiAl_2_O_4_ intermetallic phases as well as α-Al_2_O_3_ stable oxide that improved the hot corrosion resistance of nickel superalloys. It was thus demonstrated that the CVD technology with optimized parameters could be successfully applied to enhance the performance of nickel superalloys.

## Figures and Tables

**Figure 1 materials-13-03863-f001:**
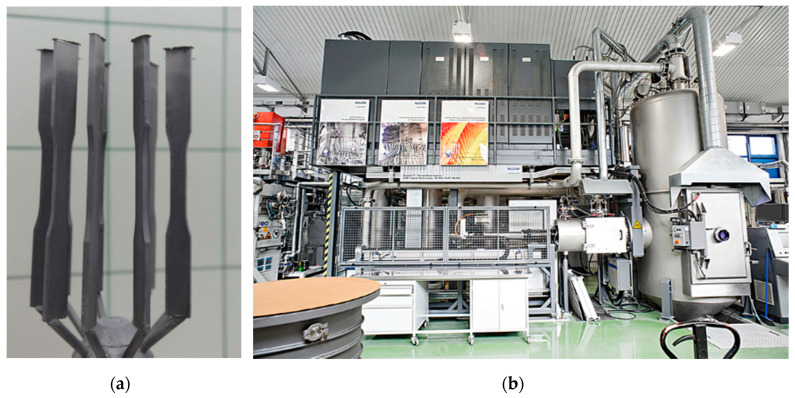
General view of MAR 247 cast samples (**a**), Ion Bond Bernex BPX Pro 325 S system for chemical vapor deposition (**b**).

**Figure 2 materials-13-03863-f002:**
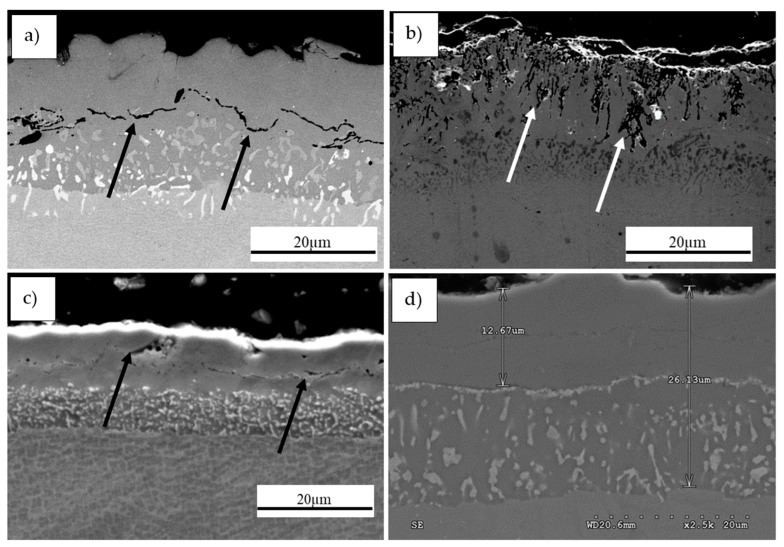
Cross-sections of intermetallic coatings produced by chemical vapor deposition (CVD) method on MAR 247 nickel superalloy at: (**a**) 880 °C for 12 h under nitrogen protective atmosphere; (**b**) 1040 °C for 8 h under nitrogen protective atmosphere; (**c**) 950 °C for 12 h under hydrogen protective atmosphere; (**d**) 1040 °C for 12 h under hydrogen protective atmosphere. Failure regions are indicated by arrows.

**Figure 3 materials-13-03863-f003:**
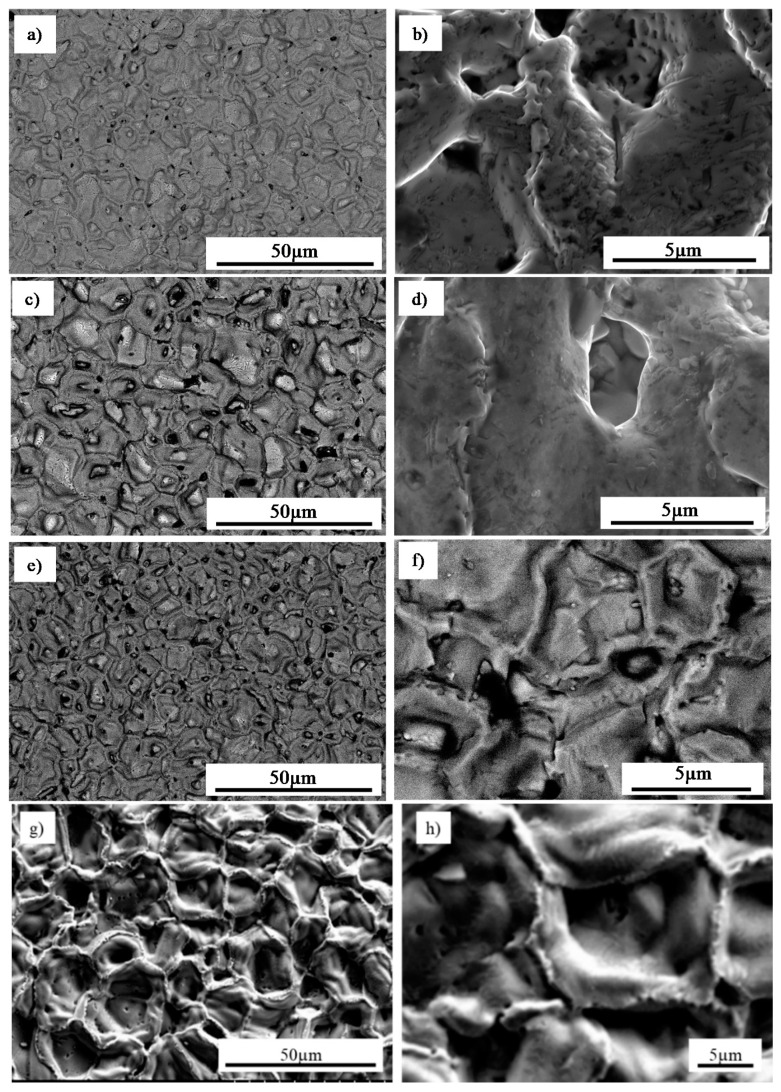
Microstructure of intermetallic coating produced by CVD method on MAR 247 nickel superalloy at 1040 °C for: (**a**,**b**) 1 h; (**c**,**d**) 2 h; (**e**,**f**) 8 h; (**g**,**h**) 12 h under hydrogen protective atmosphere.

**Figure 4 materials-13-03863-f004:**
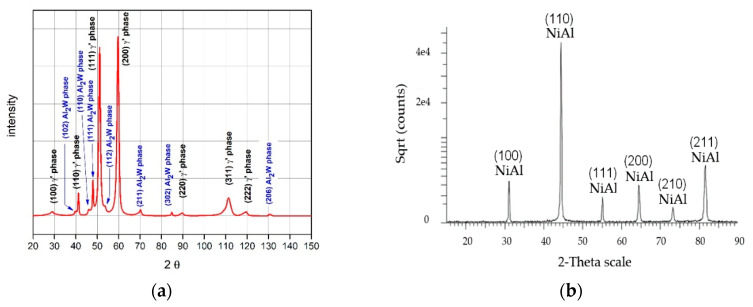
X-ray diffraction patterns from: (**a**) the as-received MAR 247 alloy without coating; (**b**) the coating produced by CVD process at the temperature of 1040 °C during 12 h of deposition under hydrogen protective atmosphere.

**Figure 5 materials-13-03863-f005:**
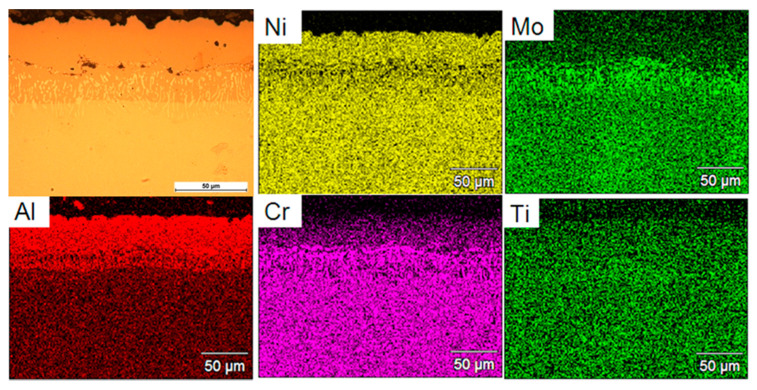
Analysis of the chemical composition from the coating surface after a low-active aluminization process at 1040 °C for 12 h.

**Figure 6 materials-13-03863-f006:**
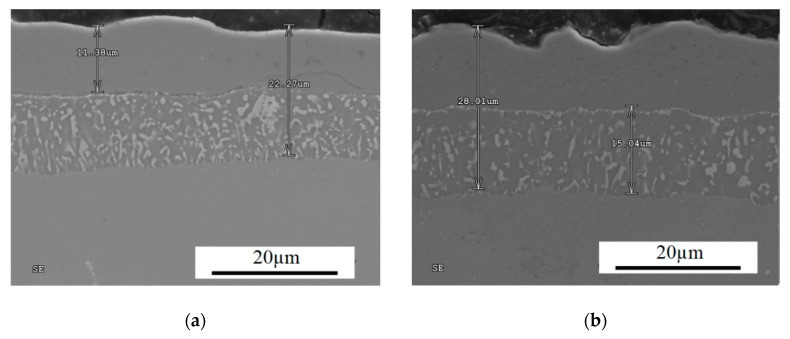
Cross-sections of aluminide coating with uniform thickness due to CVD process at the temperature of 1040 °C during 12 h of deposition. (**a**) the coating of 22.27 µm thickness; (**b**) the coating of 28.01 µm thickness.

**Figure 7 materials-13-03863-f007:**
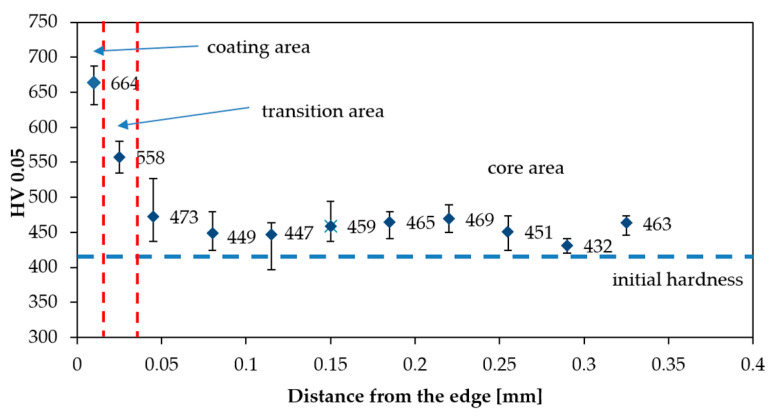
Distribution of microhardness in the cross-section of aluminized alloy.

**Figure 8 materials-13-03863-f008:**
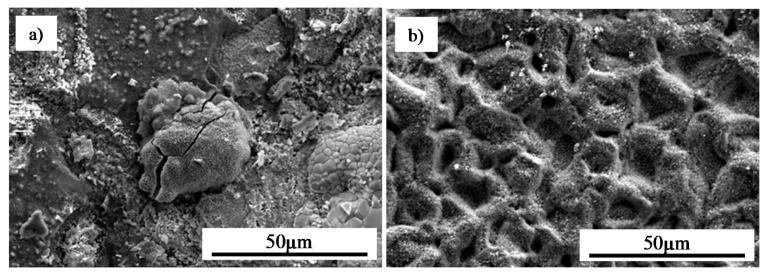
Morphology of the oxidized MAR 247 nickel superalloy: (**a**) in the initial state; (**b**) after the aluminization process.

**Figure 9 materials-13-03863-f009:**
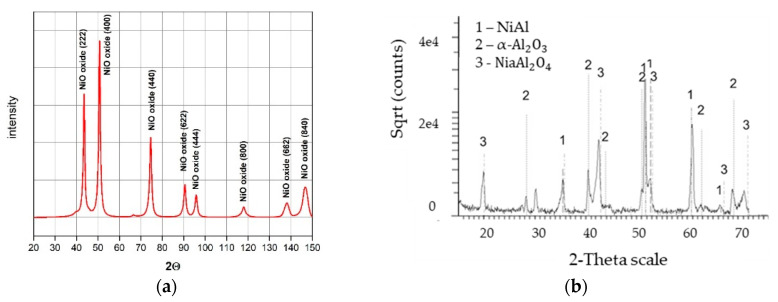
X-ray diffraction patterns determined for: (**a**) the raw MAR 247 alloy; (**b**) coated MAR 247 alloy after 24 h annealing in air conditions.

**Figure 10 materials-13-03863-f010:**
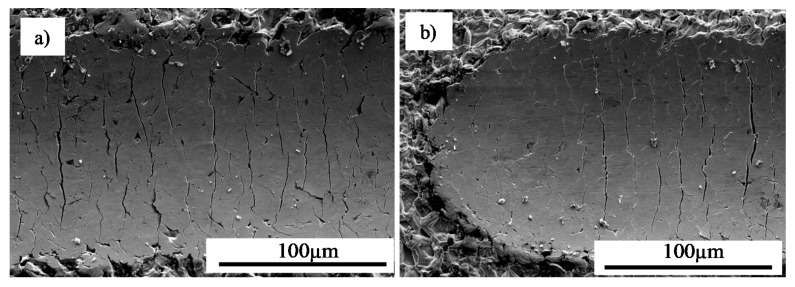
Morphology of the aluminide coating after scratch test: (**a**) central part and (**b**) scratch tip.

**Figure 11 materials-13-03863-f011:**
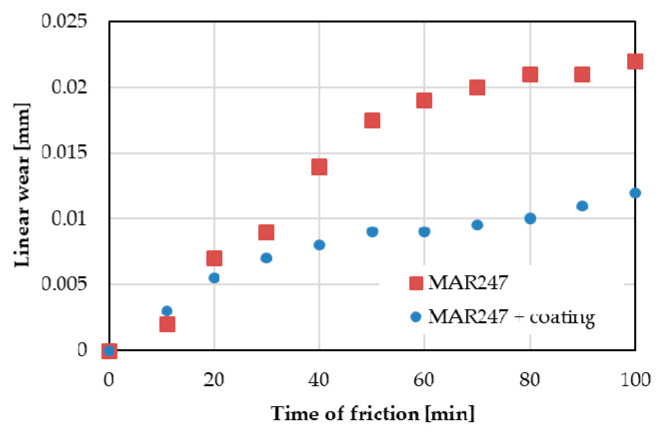
Linear wear of the as-received MAR 247 alloy without and with protective coating.

**Figure 12 materials-13-03863-f012:**
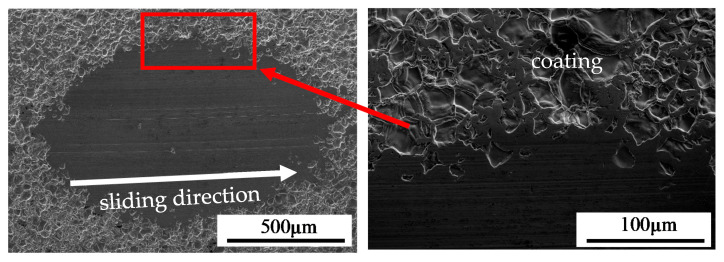
Marks of wear for MAR 247 nickel superalloy with NiAl coating.

**Table 1 materials-13-03863-t001:** Chemical composition of MAR 247 nickel superalloy (wt.%).

C	Cr	Mn	Si	W	Co	Al	Ni
0.09	8.80	0.10	0.25	9.70	9.50	5.70	bal.

**Table 2 materials-13-03863-t002:** The test matrix of CVD parameters.

Temperature [°C]	Deposition Time [h]	Protective Gas
1040	1	hydrogen
1040	2	hydrogen
1040	8	hydrogen
1040	12	hydrogen

**Table 3 materials-13-03863-t003:** Chemical EDS area composition of coating surface deposited at 1040 °C for 12 h under hydrogen protective atmosphere.

	Al	Cr	Fe	Co	Ni
at.%	42.55	0.66	0.43	4.85	51.51

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
