# Peer review of "Thermal Barrier Stability and Wear Behavior of CVD Deposited Aluminide Coatings for MAR 247 Nickel Superalloy"

_materials, 2020, doi:10.3390/ma13173863_

Round 1

Reviewer 1 Report

1) The title should be revised, emphasizing the type of the coating (aluminides). From the present title, it seems like that the type of the coating is nickel superalloy. Besides, the first letters of the nickel superalloy may not be capitalized.

2) Essential quantitative results should be provided in the abstract, such as optimized parameters.

3) Too many literatures are cited together, without enough descriptions, like [1-4], [5-7], [8-11] and [12-15], one or two key literatures is enough for explaining such one sentence.

4) Thermal Barrier Coatings/Bond Coat (TBC/BC). The first letters may not be capitalized, and the coat may be coating.

5) Only the deposition time varies in the present study, making the content little. For optimizing deposition parameters, some other parameters, like temperature and pressure, are also important. It is suggested that more studies should be added.

Author Response

Detailed Response to Reviewer Comments

Ms. Ref. No.: materials-889545

Title: Thermal barrier and wear aluminide resistant coatings for MAR 247 nickel superalloy

Materials Dear Sir or Madame,  

I would like to thank you very much for your letter and the reviewer’s comments on our manuscript (No.: materials – 889545). We appreciate your very valuable comments, that gave us a chance for revising the manuscript.

We have addressed all of the comments and revised the manuscript accordingly. All of the changes have been highlighted in yellow in the revised manuscript. Detailed responses to the comments are described in the “Response to Reviewers” point by point.

We now resubmit the manuscript for your further consideration for publication in your journal. We sincerely hope this revised manuscript will be finally acceptable for publication. If you have any questions about this manuscript, please do not hesitate to contact me.

Best regards

Mateusz Kopec

On behalf of all co-authors

Institute of Fundamental Technological Research

Polish Academy of Sciences

Reviewer’s Comments:

Reviewer #1:

1) The title should be revised, emphasizing the type of the coating (aluminides). From the present title, it seems like that the type of the coating is nickel superalloy. Besides, the first letters of the nickel superalloy may not be capitalized.

Response: We would like to thank the reviewer for the comment. The tittle was changed to “Thermal barrier stability and wear resistant behavior of CVD deposited aluminide coatings for MAR 247 nickel superalloy”.

2) Essential quantitative results should be provided in the abstract, such as optimized parameters.

Response: We would like to thank the reviewer for the comment. Corrections were highlighted in the article and detailed response is located below.

The paper reports, that the aluminides coatings of various thickness and microstructure uniformity deposited by chemical vapor deposition (CVD) process using different parameters allow to form thermal barrier and bond coating which interacting with external environment in the air atmosphere at 1100°C for 24 hours in the thermal stability test conditions. The structure and physico–chemical properties, combined with very dense and pore free aluminide coatings by optimized parameters of CVD process at 1040°C for 12 hours in protective hydrogen atmosphere were obtained. As a consequence, beneficial mechanical features including thermal stability, wear resistance and good adhesion strength to MAR 247 nickel superalloy substrate were achieved. A microstructure of the layers was characterized through the scanning electron microscopy (SEM), X-ray Energy Dispersive Spectroscopy (EDS) and X-ray diffraction (XRD) analysis. Mechanical properties and wear resistance of aluminides coatings were examined using microhardness, scratch test and standardized wear test, respectively. Intermetallic phases from the Ni-Al system of specific thickness (20-30µm), chemical and phase composition, were successfully established at optimized CVD process parameters. They were characterized by the high performance properties such as great heat, adhesion and abrasion resistance.

3) Too many literatures are cited together, without enough descriptions, like [1-4], [5-7], [8-11] and [12-15], one or two key literatures is enough for explaining such one sentence.

Response: We would like to thank the reviewer for the comment. Corrections were highlighted in the article.

4) Thermal Barrier Coatings/Bond Coat (TBC/BC). The first letters may not be capitalized, and the coat may be coating.

Response: We would like to thank the reviewer for the comment. Corrections were highlighted in the article.

5) Only the deposition time varies in the present study, making the content little. For optimizing deposition parameters, some other parameters, like temperature and pressure, are also important. It is suggested that more studies should be added.

Response: We would like to thank the reviewer for the comment. Corrections were highlighted in the article.

The data matrix of CVD parameters used throughout the studies was presented in Table 2. Temperature of 1040°C and two different protective gases were applied for deposition time ranging from 1 – 12 hours to assess the maximum effectiveness of the process. These parameters were selected on the basis of preliminary studies carried out in different temperatures from the range of 880°C to 1040°C. It was found that CVD process performed at lower temperatures (880°C, 950°C) with a relatively long time of deposition (up to 12 hours) and different protective atmosphere did not allow to obtain a non-defected and cohesive coatings as presented in the samples cross-section (Figure 2).

The deposition process performed at the initial temperature of 880°C for 12 hours under nitrogen protective atmosphere results in the incoherent coating with visible cracks along the surface (Figure 2a). Subsequent increase of process temperature to 1040°C led to excessive deterioration of coating obtained, Figure 2b. Based on such observations, authors decide to not extend the deposition time for nitrogen protective atmosphere. The same conditions were used for the hydrogen atmosphere. The temperature less than 950°C did not allow to obtain a non-defected coating as cracks were still found within the structure (Figure 2c). Based on the microstructural observations, it was found, that temperature of 1040°C and hydrogen as protective gas were sufficient to perform the CVD process since the aluminide coating was obtained successfully (Figure 2d), and thus, the different deposition time was further studied in details.

Figure 2. Cross-sections of intermetallic coatings produced by CVD method on MAR 247 nickel superalloy at: (a) 880°C for 12 hours under nitrogen protective atmosphere; (b) 1040°C for 8 hours under nitrogen protective atmosphere; (c) 950°C for 12 hours under hydrogen protective atmosphere; (d) 1040°C for 12 hours under hydrogen protective atmosphere.

Reviewer 2 Report

The manuscript started from a series of CVD processes, from which an ‘optimal’ condition was identified, followed by a range of microstructure and micromechanics testing. Overall, the manuscript is well written and prepared. However, the reviewer still has questions that hope authors would address first.

  1. Table 2 summarised a list of chosen parameters. Here more information is required on why, namely rationale of choosing these parameters. Without clear explanation, the reviewer has many questions, e.g. why 1040 C? Why not other temperature range? Why only 1, 2, 8, 12 but not beyond? Why nitrogen was only chosen for 8 hours, as it might work perfectly well for 12 and beyond? The reviewer thinks this is critical, as all follow-up characterisations were based on the above criterion and the key of the manuscript is ‘optimization’ as highlighted in the title. So, this needs to be treated carefully.
  2. It seems that the criterion of choosing the best processed sample was primarily based on ‘non-defected structure’ from the SEM observation. The SEM is clearly a surface sensitive technique. How would authors justify there would be no voids/defects inside the samples? Could authors consider showing different cross-sectional areas to confirm if CT or/and FIB-SEM tomography are not possible and available? Same as above, this is critical as the following characterisations were based on the criterion.
  3. If reviewer is correct, from Figure 3, the characterisations were focused on ‘1040 C for 12 hours’ only. It would be better to make this clear in the texts or by adding clearer sub-title. It was a bit confusing after Figure 1 that, the reviewer thought the following characterisations covered all different samples.
  4. Could authors explain why these specific heat resistance testing parameters were chosen: ‘1100 C for 24 hours’? Any rationale or/and references/standards that authors can refer to.

Author Response

Detailed Response to Reviewer Comments

Ms. Ref. No.: materials-889545

Title: Thermal barrier and wear aluminide resistant coatings for MAR 247 nickel superalloy

Materials Dear Sir or Madame,  

I would like to thank you very much for your letter and the reviewer’s comments on our manuscript (No.: materials – 889545). We appreciate your very valuable comments, that gave us a chance for revising the manuscript.

We have addressed all of the comments and revised the manuscript accordingly. All of the changes have been highlighted in yellow in the revised manuscript. Detailed responses to the comments are described in the “Response to Reviewers” point by point.

We now resubmit the manuscript for your further consideration for publication in your journal. We sincerely hope this revised manuscript will be finally acceptable for publication. If you have any questions about this manuscript, please do not hesitate to contact me.

Best regards

Mateusz Kopec

On behalf of all co-authors

Institute of Fundamental Technological Research

Polish Academy of Sciences

Reviewer’s Comments:
Reviewer #2

The manuscript started from a series of CVD processes, from which an ‘optimal’ condition was identified, followed by a range of microstructure and micromechanics testing. Overall, the manuscript is well written and prepared. However, the reviewer still has questions that hope authors would address first.

1.Table 2 summarized a list of chosen parameters. Here more information is required on why, namely rationale of choosing these parameters. Without clear explanation, the reviewer has many questions, e.g. why 1040 C? Why not other temperature range? Why only 1, 2, 8, 12 but not beyond? Why nitrogen was only chosen for 8 hours, as it might work perfectly well for 12 and beyond? The reviewer thinks this is critical, as all follow-up characterizations were based on the above criterion and the key of the manuscript is ‘optimization’ as highlighted in the title. So, this needs to be treated carefully.

Response: We would like to thank the reviewer for the comment. Corrections were highlighted in the article and presented below.

The data matrix of CVD parameters used throughout the studies was presented in Table 2. Temperature of 1040°C and two different protective gases were applied for deposition time ranging from 1 - 12 hours to assess the maximum effectiveness of the process. These parameters were selected on the basis of preliminary studies were different temperature range (880°C - 1040°C) was used. It was found that CVD process performed at lower temperatures (880°C, 950°C) with relatively long time of deposition (up to 12 hours) and different protective atmosphere did not allow to obtain a non-defected and cohesive coatings as presented in the samples cross-section (Fig.1).

The deposition process performed at the initial temperature of 880°C for 12 hours under nitrogen protective atmosphere results in incoherent coating with visible cracks along the surface (Figure 2a). Subsequent increase of process temperature to 1040°C led to excessive deterioration of coating obtained as observed in Figure 2b. Based on such observations, authors decide to not extend the deposition time for nitrogen protective atmosphere. The same conditions were used for the hydrogen atmosphere. The temperature up to 950°C did not allow to obtain a non-defected coating as cracks were still found within the structure (Figure 2c). Based on the microstructural observations, it was found, that temperature of 1040°C and hydrogen as protective gas were sufficient to perform the CVD process as aluminide coating was obtained successfully (Figure 2d) and thus the different deposition time was further studied in details.

Figure 2. Cross-sections of intermetallic coatings produced by CVD method on MAR 247 nickel superalloy at 880°C for 12 hours under nitrogen protective atmosphere (a), at 1040°C for 8 hours under nitrogen protective atmosphere (b), at 950°C for 12 hours under hydrogen protective atmosphere (c), at 1040°C for 12 hours under hydrogen protective atmosphere (d).           

2.It seems that the criterion of choosing the best processed sample was primarily based on ‘non-defected structure’ from the SEM observation. The SEM is clearly a surface sensitive technique. How would authors justify there would be no voids/defects inside the samples? Could authors consider showing different cross-sectional areas to confirm if CT or/and FIB-SEM tomography are not possible and available? Same as above, this is critical as the following characterisations were based on the criterion.

Response: We would like to thank the reviewer for the comment. Corrections were highlighted in the article. Different cross-sections were presented in Figure 2.

3.If reviewer is correct, from Figure 3, the characterisations were focused on ‘1040 C for 12 hours’ only. It would be better to make this clear in the texts or by adding clearer sub-title. It was a bit confusing after Figure 1 that, the reviewer thought the following characterisations covered all different samples.

Response: We would like to thank the reviewer for the comment. Corrections were highlighted in the article.

4.Could authors explain why these specific heat resistance testing parameters were chosen: ‘1100 C for 24 hours’? Any rationale or/and references/standards that authors can refer to.

Response: We would like to thank the reviewer for the comment. Corrections were highlighted in the article and presented below.

The extreme performance conditions of modern aircraft engine turbines require the application of heat-resistant materials. The maximum operating temperature of contemporary nickel superalloys is 1100˚C which is why it is necessary to use a protective coatings on the hot parts of the aircraft engine turbines [16]. Thus the heat resistance test of MAR 247 alloy with aluminide protective coating was performed in the air atmosphere at 1100°C for 24 hours.

  1. Mori, T., Kuroda, S., Murakami, H., Katanoda, H., Sakamoto, Y., Newman, S., Effects of initial oxidation on beta phase depletion and oxidation of CoNiCrAlY bond coatings fabricated by warm spray and HVOF processes. Sufrace & Coatings Technology 221 (2013) 59-69.

Reviewer 3 Report

This manuscript presents the results of investigating the microstructure, mechanical and wear properties of aluminide coatings deposited by the CVD process on nickel superalloy.

Since the title and the first sentence of the Abstract should be connected, I suggest mentioning nickel superalloys in the first sentence of the Abstract i.e. not to use only aluminide coatings but two terms (nickel superalloys and aluminide coatings). On the other hand, the authors can include aluminide coatings term into the title (for instance, “… wear resistant aluminide coatings”).

The subject of the investigation is not primarily the optimization of CVD parameters, and therefore, I suggest changing the title (for example, “Microstructural and mechanical properties of …”). When it is about optimization of some process, then some techniques (e.g. design of experiment combined with genetic algorithm) should be used and optimal parameters obtained. When I read the title for the first time, I was expecting something like that.

Please cite figures and tables in the text as Figure 1, Table 1 (not Fig. 1 or Tab. 1; especially not e.g. Fig.1, Tab.3 - without space).  

Please use the same indentation of the paragraphs.

Please, use uniformly the same term (MAR247 or MAR 247).

I suggest inserting some photos of samples and equipment used. Moreover, I would like to suggest to the authors to explain in detail the number of samples used for their valuable investigation.

Line 77 – „to achieve“ instead of „to achieved“

Lines 106, 107 – Please, check if this is correct (English language) – „a typical examples“.

I would like to propose moving the sentence „Since the nitrogen atmosphere was used as the carrier gas, a more porous layer was observed (Fig. 1i-j).”, after the sentence „A considerable change of the morphology can be obtained when the deposition time was extended up to 8 hours (Fig.1e-h).“ to finish explaining Figure 1. Furthermore, the atmosphere should be mentioned when explaining the Figure 1e-h.

Please, do not stick the numbers and measurement units (8hours, 12h, 50g, 664HV0.05, etc.).

Please use uniformly, i.e. use one term: h, hours; sublayer, sub-layer; 9.96% at., 50 at.%, 55.4% at; 664HV0.05, 650 HV; aluminide coating, aluminides layer, aluminized alloy…

Line 123 – “Since the nitrogen atmosphere was used as the carrier gas…“ – maybe it would be better to use the following „When the nitrogen atmosphere was used as the carrier gas…“ – please consult a native English speaker.

The title of Table 3 should be more detailed (for which parameters).

Low-active aluminisation (LAHT) – maybe low-active high-temperature aluminisation (LAHT).

„The EDX map analysis also confirmed the assumptions of LAHT aluminisation (900–1150°C) process reported in literature [22].“ – In this sentence, Figure 3 should be cited (e.g. „The EDX map analysis (Figure 3)…“).

 “Distance from the edge [mm]” instead of “Distance from the edge[mm]”.

Since the authors investigated heat resistance of the coatings as well, it should be mentioned in the Abstract.

The last sentence of the Abstract is as follows: „They were characterized by high performance properties such as great adhesion and abrasion resistance.“ Here, the abrasion resistance is mentioned (and in the section 2 as well), but in the sub-section 3.4, the sliding wear is mentioned. Was it about abrasive or sliding wear?

Conclusion is too general. It has to be more detailed with provided highlights and the scientific contribution of the investigation conducted.

The references should be more careful written (according to the Template). There are many errors.

Author Response

Detailed Response to Reviewer Comments

Ms. Ref. No.: materials-889545

Title: Thermal barrier and wear aluminide resistant coatings for MAR 247 nickel superalloy

Materials Dear Sir or Madame,  

I would like to thank you very much for your letter and the reviewer’s comments on our manuscript (No.: materials – 889545). We appreciate your very valuable comments, that gave us a chance for revising the manuscript.

We have addressed all of the comments and revised the manuscript accordingly. All of the changes have been highlighted in yellow in the revised manuscript. Detailed responses to the comments are described in the “Response to Reviewers” point by point.

We now resubmit the manuscript for your further consideration for publication in your journal. We sincerely hope this revised manuscript will be finally acceptable for publication. If you have any questions about this manuscript, please do not hesitate to contact me.

Best regards

Mateusz Kopec

On behalf of all co-authors

Institute of Fundamental Technological Research

Polish Academy of Sciences

Reviewer’s Comments:

Reviewer #3

This manuscript presents the results of investigating the microstructure, mechanical and wear properties of aluminide coatings deposited by the CVD process on nickel superalloy.

1.Since the title and the first sentence of the Abstract should be connected, I suggest mentioning nickel superalloys in the first sentence of the Abstract i.e. not to use only aluminide coatings but two terms (nickel superalloys and aluminide coatings). On the other hand, the authors can include aluminide coatings term into the title (for instance, “… wear resistant aluminide coatings”).

Response: We would like to thank the reviewer for the comment. Corrections were highlighted in the article and presented below. The title was changed to “Thermal barrier stability and wear resistant behavior of CVD deposited aluminide coatings for MAR 247 nickel superalloy” and the abstract was rewritten.

The paper reports, that the aluminides coatings of various thickness and microstructure uniformity deposited by chemical vapor deposition (CVD) process using different parameters allow to form thermal barrier and bond coating which interacting with external environment in the air atmosphere at 1100°C for 24 hours in the thermal stability test conditions. The structure and physico–chemical properties, combined with very dense and pore free aluminide coatings by optimized parameters of CVD process at 1040°C for 12 hours in protective hydrogen atmosphere were obtained. As a consequence, beneficial mechanical features including thermal stability, wear resistance and good adhesion strength to MAR 247 nickel superalloy substrate were achieved. A microstructure of the layers was characterized through the scanning electron microscopy (SEM), X-ray Energy Dispersive Spectroscopy (EDS) and X-ray diffraction (XRD) analysis. Mechanical properties and wear resistance of aluminides coatings were examined using microhardness, scratch test and standardized wear test, respectively. Intermetallic phases from the Ni-Al system of specific thickness (20-30µm), chemical and phase composition, were successfully established at optimized CVD process parameters. They were characterized by the high performance properties such as great heat, adhesion and abrasion resistance.

2.The subject of the investigation is not primarily the optimization of CVD parameters, and therefore, I suggest changing the title (for example, “Microstructural and mechanical properties of …”). When it is about optimization of some process, then some techniques (e.g. design of experiment combined with genetic algorithm) should be used and optimal parameters obtained. When I read the title for the first time, I was expecting something like that.

Response: We would like to thank the reviewer for the comment. Corrections were highlighted in the article and presented below. The title was changed to “Thermal barrier stability and wear resistant behavior of CVD deposited aluminide coatings for MAR 247 nickel superalloy”

Please cite figures and tables in the text as Figure 1, Table 1 (not Fig. 1 or Tab. 1; especially not e.g. Fig.1, Tab.3 - without space).  

Response: We would like to thank the reviewer for the comment. Corrections were made accordingly.

3.Please use the same indentation of the paragraphs.

Response: We would like to thank the reviewer for the comment. Corrections were made accordingly.

4.Please, use uniformly the same term (MAR247 or MAR 247).

Response: We would like to thank the reviewer for the comment. Corrections were made accordingly.

5.I suggest inserting some photos of samples and equipment used. Moreover, I would like to suggest to the authors to explain in detail the number of samples used for their valuable investigation.

Response: We would like to thank the reviewer for the comment. Corrections were made accordingly.

6.Line 77 – „to achieve“ instead of „to achieved“

7.Lines 106, 107 – Please, check if this is correct (English language) – „a typical examples“.

Response: We would like to thank the reviewer for the comment. Corrections were made accordingly.

8.I would like to propose moving the sentence „Since the nitrogen atmosphere was used as the carrier gas, a more porous layer was observed (Fig. 1i-j).”, after the sentence „A considerable change of the morphology can be obtained when the deposition time was extended up to 8 hours (Fig.1e-h).“ to finish explaining Figure 1. Furthermore, the atmosphere should be mentioned when explaining the Figure 1e-h.

Response: We would like to thank the reviewer for the comment. Since the nitrogen did not allows to obtain a non-defected coating, the coating obtained using this atmosphere was not included as explained in lines 114-124.

9.Please, do not stick the numbers and measurement units (8hours, 12h, 50g, 664HV0.05, etc.).

10.Please use uniformly, i.e. use one term: h, hours; sublayer, sub-layer; 9.96% at., 50 at.%, 55.4% at; 664HV0.05, 650 HV; aluminide coating, aluminides layer, aluminized alloy…

Response: We would like to thank the reviewer for the comment. Corrections were made accordingly.

11.Line 123 – “Since the nitrogen atmosphere was used as the carrier gas…“ – maybe it would be better to use the following „When the nitrogen atmosphere was used as the carrier gas…“ – please consult a native English speaker.

The title of Table 3 should be more detailed (for which parameters).

Low-active aluminisation (LAHT) – maybe low-active high-temperature aluminisation (LAHT).

Response: We would like to thank the reviewer for the comment. Corrections were made accordingly.

12.„The EDX map analysis also confirmed the assumptions of LAHT aluminisation (900–1150°C) process reported in literature [22].“ – In this sentence, Figure 3 should be cited (e.g. „The EDX map analysis (Figure 3)…“).

 “Distance from the edge [mm]” instead of “Distance from the edge[mm]”.

Response: We would like to thank the reviewer for the comment. Corrections were made accordingly.

13.Since the authors investigated heat resistance of the coatings as well, it should be mentioned in the Abstract.

Response: We would like to thank the reviewer for the comment. Corrections were made accordingly.

14.The last sentence of the Abstract is as follows: „They were characterized by high performance properties such as great adhesion and abrasion resistance.“ Here, the abrasion resistance is mentioned (and in the section 2 as well), but in the sub-section 3.4, the sliding wear is mentioned. Was it about abrasive or sliding wear?

Response: We would like to thank the reviewer for the comment. The wear resistance investigations was made regarding PN-83/H-04302 using T-04 device designed to assess the wear resistance of coating materials used for sliding elements of machines. The test association (Figure A) consists of three stationary rollers with deposited CVD coatings, spaced at 120º and pressed with 200 MPa load against a conical counter-specimen made of the SW7W high-speed steel and rotating for 100 minutes at a speed of 160 rpm. At the concentrated point of contact, the movement of the conical counter-sample with respect to the stationary rollers is sliding.

Figure A. Schema of T-04 device for wear resistance tests.

15.Conclusion is too general. It has to be more detailed with provided highlights and the scientific contribution of the investigation conducted.

Response: We would like to thank the reviewer for the comment. Corrections were made accordingly.

Chemical vapor deposition process for MAR 247 nickel superalloy at the temperature of 1040°C kept for 12 hours in the protective hydrogen atmosphere allowed to obtain a non-defected substrate material with thermal barrier and wear resistant NiAl coatings of uniform thickness ranging from 20µm to 30 µm. The coating was characterized by the very good adherence, wear and thermal resistance confirmed in studies performed. Its application improved the mechanical properties such as hardness and wear almost twice than that of the as-received MAR 247 alloy. The coating hardness of 664 HV0.05 was associated with the distribution of the aluminum atoms in the lattice of NiAl intermetallic coating formed during high temperature deposition.  The aluminide coatings exhibited a very good adherence during scratch tests as the breakdown of the layer was not observed on the coating surface after tests. The oxidized aluminide coating was characterized by the excellent durability and tightness of the protective scale as no scale spallation was observed. The phase structure of scale consisted mainly of NiAl and NiAl2O4 intermetallic phases as well as a-Al2O3 stable oxide that improve the hot corrosion resistance of nickel superalloy. The CVD technology with parameters determined could be successfully applied to enhance the different useful properties of nickel superalloys.

16.The references should be more careful written (according to the Template). There are many errors.

Response: We would like to thank the reviewer for the comment. Corrections were made accordingly.

Round 2

Reviewer 1 Report

n/a

Author Response

Authors would like to thank for valuable comments and suggestions.

Reviewer 3 Report

The authors did a great effort to improve the presentation of their valuable investigation. However, the authors are encouraged to take into account the following suggestions and comments:

Title – “wear resistance” instead of “wear resistant” (or the word “resist…” can be avoided) - – please consult a native English speaker.

Abstract – „which interacting“ – „which interact“ or „interacting“ – please consult a native English speaker.

Abstract – The first sentence is too long and complicated. The second sentence is very hard to understand. In general, the Abstract is now too complicated and difficult to understand and does not summarize the topic of investigation in appropriate way.

Please, do not stick the numbers and measurement units 30μm, etc.

In the revised version, authors omitted the condition “1040, 8, nitrogen gas”, from Table 2, i.e. they continue their manuscript with only one protective gas (hydrogen). Accordingly, they should be very careful because it is reflected on the following sentences:

„The data matrix of CVD parameters used throughout the studies was presented in Table 2. Temperature of 1040°C and two different protective gases were applied for deposition time ranging from 1 – 12 hours to assess the maximum effectiveness of the process.“

„Microstructural observations revealed that protective atmosphere affects the coating morphology.“

Figures 3 g and h present the time of 12 hours, and the following sentence should be modified: „A considerable change of the morphology can be obtained when the deposition time was extended up to 8 hours (Figure 3e-h).“ Hence, the Figures 3 g and h should be described.

The authors should choose between EDS and EDX; sublayer and sub-layer, aluminum and aluminium, etc. (please use one term).

„…with a thickness of about 35 μm consists mainly aluminum…“ – suggestion: „contains“ instead of „consists“ – please consult  a native English speaker.

„The abrasive wear test was carried out using the three rollers + cone method in accordance to the Polish Standard (PN-83 / 101 H04302).“  - According to the explanation of authors in coverletter, the word abrasive should be changed to sliding.

References – uniformity is needed (e.g. with doi, with the abbreviated titles of the journals, etc.)

Suggestion for further manuscripts – the section (chapter) should not end with table or figure, but with paragraph (text).

Author Response

The authors did a great effort to improve the presentation of their valuable investigation. However, the authors are encouraged to take into account the following suggestions and comments:

Title – “wear resistance” instead of “wear resistant” (or the word “resist…” can be avoided) - – please consult a native English speaker.

Authors would like to thank for the comment. Word “resist” was avoided.

Abstract – „which interacting“ – „which interact“ or „interacting“ – please consult a native English speaker.

Authors would like to thank for the comment. Corrections were highlighted in the article using green colour.

Abstract – The first sentence is too long and complicated. The second sentence is very hard to understand. In general, the Abstract is now too complicated and difficult to understand and does not summarize the topic of investigation in appropriate way.

In this paper the aluminides coatings of various thickness and microstructure uniformity obtained using chemical vapor deposition (CVD) process and different parameters were studied in details. The optimized parameters of CVD process at 1040°C for 12 hours in protective hydrogen atmosphere allow to obtain a very dense and pore free aluminide coatings. These coatings were characterized by beneficial mechanical features including thermal stability, wear resistance and good adhesion strength to MAR 247 nickel superalloy substrate. A microstructure of the layers was characterized through the scanning electron microscopy (SEM), X-ray Energy Dispersive Spectroscopy (EDS) and X-ray diffraction (XRD) analysis. Mechanical properties and wear resistance of aluminides coatings were examined using microhardness, scratch test and standardized wear test, respectively. Intermetallic phases from the Ni-Al system of specific thickness (20 - 30 µm), chemical and phase composition were successfully established at optimized CVD process parameters. They were characterized by the high performance properties such as great heat, adhesion and abrasion resistance.

Please, do not stick the numbers and measurement units 30μm, etc.

Authors would like to thank for the comment. Corrections were highlighted in the article using green colour.

In the revised version, authors omitted the condition “1040, 8, nitrogen gas”, from Table 2, i.e. they continue their manuscript with only one protective gas (hydrogen). Accordingly, they should be very careful because it is reflected on the following sentences:

„The data matrix of CVD parameters used throughout the studies was presented in Table 2. Temperature of 1040°C and two different protective gases were applied for deposition time ranging from 1 – 12 hours to assess the maximum effectiveness of the process.“

„Microstructural observations revealed that protective atmosphere affects the coating morphology.“

Authors would like to thank for the comment. Corrections were highlighted in the article using green colour.

Figures 3 g and h present the time of 12 hours, and the following sentence should be modified: „A considerable change of the morphology can be obtained when the deposition time was extended up to 8 hours (Figure 3e-h).“ Hence, the Figures 3 g and h should be described.

 Authors would like to thank for the comment. Corrections were highlighted in the article using green colour.

The authors should choose between EDS and EDX; sublayer and sub-layer, aluminum and aluminium, etc. (please use one term).

Authors would like to thank for the comment. Corrections were highlighted in the article using green colour.

„…with a thickness of about 35 μm consists mainly aluminum…“ – suggestion: „contains“ instead of „consists“ – please consult  a native English speaker.

Authors would like to thank for the comment. Corrections were highlighted in the article using green colour.

„The abrasive wear test was carried out using the three rollers + cone method in accordance to the Polish Standard (PN-83 / 101 H04302).“  - According to the explanation of authors in coverletter, the word abrasive should be changed to sliding.

Authors would like to thank for the comment. Corrections were highlighted in the article using green colour.

References – uniformity is needed (e.g. with doi, with the abbreviated titles of the journals, etc.)

Authors would like to thank for the comment. Corrections were made accordingly.

Suggestion for further manuscripts – the section (chapter) should not end with table or figure, but with paragraph (text).

Authors would like to thank for the comment.